# Genomic Insights into *Pseudomonas protegens* E1BL2 from Giant Jala Maize: A Novel Bioresource for Sustainable Agriculture and Efficient Management of Fungal Phytopathogens

**DOI:** 10.3390/ijms25179508

**Published:** 2024-09-01

**Authors:** Esaú De la Vega-Camarillo, Josimar Sotelo-Aguilar, Adilene González-Silva, Juan Alfredo Hernández-García, Yuridia Mercado-Flores, Lourdes Villa-Tanaca, César Hernández-Rodríguez

**Affiliations:** 1Departamento de Microbiología, Escuela Nacional de Ciencias Biológicas, Instituto Politécnico Nacional, Prolongación de Carpio y Plan de Ayala s/n, Col. Santo Tomás, Ciudad de México 11340, Mexico; edelavegac1000@alumno.ipn.mx (E.D.l.V.-C.); osoteloa1300@alumno.ipn.mx (J.S.-A.); adigonzalezs@ipn.mx (A.G.-S.); jahernandezga@ipn.mx (J.A.H.-G.); mvillat@ipn.mx (L.V.-T.); 2Laboratorio de Aprovechamiento Integral de Recursos Bióticos, Universidad Politécnica de Pachuca, Carretera Pachuca-Ciudad Sahagún Km. 20, Rancho Luna, Ex-Hacienda de Santa Bárbara Zempoala, Pachuca 43830, Mexico; yuridiamercado@upp.edu.mx

**Keywords:** *Pseudomonas protegens*, bacterial genome, biocontrol, phytopathogen, maize endophyte, promotion growth plant, antifungals

## Abstract

The relationships between plants and bacteria are essential in agroecosystems and bioinoculant development. The leaf endophytic *Pseudomonas protegens* E1BL2 was previously isolated from giant Jala maize, which is a native *Zea mays* landrace of Nayarit, Mexico. Using different Mexican maize landraces, this work evaluated the strain’s plant growth promotion and biocontrol against eight phytopathogenic fungi in vitro and greenhouse conditions. Also, a plant field trial was conducted on irrigated fields using the hybrid maize Supremo. The grain productivity in this assay increased compared with the control treatment. The genome analysis of *P. protegens* E1BL2 showed putative genes involved in metabolite synthesis that facilitated its beneficial roles in plant health and environmental adaptation (*bdhA*, *acoR*, *trpE*, *speE*, *potA*); siderophores (*ptaA*, *pchC*); and extracellular enzymes relevant for PGPB mechanisms (*cel3*, *chi14*), protection against oxidative stress (*hscA*, *htpG*), nitrogen metabolism (*nirD*, *nit1*, *hmpA*), inductors of plant-induced systemic resistance (ISR) (*flaA*, *flaG*, *rffA*, *rfaP*), fungal biocontrol (*phlD*, *prtD*, *prnD*, *hcnA-1*), pest control (*vgrG-1*, *higB-2*, *aprE*, *pslA*, *ppkA*), and the establishment of plant-bacteria symbiosis (*pgaA*, *pgaB*, *pgaC*, *exbD*). Our findings suggest that *P. protegens* E1BL2 significantly promotes maize growth and offers biocontrol benefits, which highlights its potential as a bioinoculant.

## 1. Introduction

In the current biotechnological era, there is an ongoing effort to improve agricultural practices. The focus is on moving away from external agrochemicals and broad-spectrum antimicrobials. Although these substances protect against pests and pathogens, they have also caused soil erosion, the pollution of runoff waters, and modified plant microbiomes that displace beneficial rhizospheric and endophytic microorganisms. Some of these microorganisms are crucial for promoting plant growth and protecting plant tissues by inhibiting phytopathogen proliferation [1]. This underscores the critical role of the plant microbiome, which is a key player in creating a sustainable and “eco-friendly” agroecosystem for the soil [2].

In the plant–microorganism relationship, the soil has a significant influence because a good number of microorganisms that are part of the plants’ endophytic and rhizospheric microbiome come from the bulk soil, where through ecological competition and colonization, they manage to establish a close symbiosis with the plants [3]. Many of the plant-growth-promoting bacteria (PGPB) also enhance plant health through various mechanisms. These include the production of plant hormones, like indole acetic acid (IAA), which aids in root development, and ACC deaminase (ACCd), which is crucial for stress mitigation by lowering ethylene levels, while metallophores and phosphate solubilization enhance nutrient uptake. Additionally, PGPB produces enzymes, such as cellulases and amylases, which facilitate nutrient release from organic matter in the soil [4]. Furthermore, they synthesize secondary metabolites, such as 2,4-diacetyl phloroglucinol (DAPG), pyoverdine (PV), pyrrolnitrin (PR), pyoluteorin (PL), pyocyanin (PC), orphamide A, hydrogen cyanide (HCN), and chitinases, which act as a biocontrol mechanism [5]. Among PGPB, numerous species belonging to the g-proteobacteria soil and plant-associated genera, such as *Pseudomonas*, stand out as highly significant. This significance stems from their remarkable biological versatility, which arises from the diverse genetic, physiological, and metabolic composition embedded within their genomic context [6,7,8,9].

In recent decades, many strains of *Pseudomonas* were isolated from water, soil, air, animals, human sources, and plants [10]. Thanks to isolation and molecular detection efforts, a large number of mutualistic or pathogenic species associated with plants were identified or detected, including both mutualistic and pathogenic species. Mutualistic species, including *Pseudomonas aeruginosa*, *Pseudomonas fluorescens*, *Pseudomonas syringae*, *Pseudomonas phaseolicola*, *Pseudomonas chlororaphis*, *Pseudomonas putida*, and *Pseudomonas protegens*, engage in beneficial symbiotic relationships with plants. These bacteria contribute significantly to plant growth and health promotion through mechanisms such as nutrient cycling, disease suppression, and hormone production, thereby enhancing the overall resilience and productivity of plant ecosystems [11,12,13,14]. Some species, such as *Pseudomonas stutzeri* A1501 in rice [15], *Pseudomonas putida* 2 in sesame [16], *P. fluorescens* PICF7 in barley [17], *Pseudomonas synxantha* 2–79 in canola [18], and *Pseudomonas protegens* DA1.2 in wheat [19], were highlighted as plant growth promoters and exert some biocontrol. *P. fluorescens* is a recognized PGPB that produces plant-promoting metabolites and enzymes but also inhibits the development of fungal phytopathogen species of *Fusarium*, *Aspergillus*, *Pythium*, and *Rhizoctonia* genus. *P. fluorescens* excretes antifungal compounds (PV, HCN, and DAPG). Also, these bacteria secrete some hydrolytic chitinase enzymes and superoxide dismutase (SOD), catalase (CAT), and peroxidase (PO) antioxidant enzymes [18], all of which contribute to the survival of the plant under conditions of biotic and abiotic stress [20,21]. 

A group of fluorescent *Pseudomonas* has been linked to maize plants (*Zea mays*) because they were proven to have great potential for agricultural production. The importance of this partnership lies in the future of bioinoculant output, which refers to the production of beneficial microorganisms for agricultural use [22]. The two species most widely related and described as PGPB in symbiosis with maize are *P. fluorescens* and *P. protegens*, which share a remarkable genomic and phenotypic similarity that has sometimes been indistinctly named. In particular, *P. protegens* displays exceptional plant growth promotion; enhanced nutrient uptake capabilities; and the production of antibiotics, metallophores, and enzymes that suppress plant pathogens. This species is crucial in improving crop resilience and productivity, which emphasizes its role in sustainable agricultural practices [23].

This work analyzed the phenotypical and genotypical plant-promoting features of the endophytic bacterium *P. protegens* E1BL2 previously isolated from the Jala landrace’s Mexican maize grains. The draft genome harbored many recognized genes associated with strategies to promote plant growth and suppress the development of phytopathogenic fungi. These strategies were tested in vitro, where the bacteria were grown in a controlled environment, and in planta, where they were introduced into live plants. The results showed significant growth promotion in both settings. Additionally, crop yield trials performed with native Mexican maize landraces demonstrated a substantial increase in yield, which further confirmed the bacteria’s effectiveness. Fungal biocontrol tests using common maize phytopathogen fungi showed promising results. 

## 2. Results

### 2.1. Genome Structure of P. protegens E1BL2

The genome of *P. protegens* E1BL2 consisted of a circular chromosome of 7,091,943 bp with 63.3% GC and encoded for 6,543 protein sequences. The average nucleotide identity was calculated with BLAST algorithm (ANIb) values to confirm the species’ taxonomic assignation; this gave a 99.3% similarity percentage with *P. protegens* SN15-2 (Figure 1).

### 2.2. Phylogenomic and Comparative Genomics of Stain E1BL2

The phylogenomic tree was constructed by comparing a core of 937 orthologous genes from 39 strains belonging to the genus *Pseudomonas*. Seven main clades could be observed, within which the *P. protegens* strains formed a monophyletic group closely related to *P. chlororaphis* (Figure 2).

In general, comparative genomics of *P. protegens* revealed that genomes of all strains were enriched with gene pathways to the biosynthesis of phytohormones, fungal biological control, cellulases, quorum sensing, quenching, catalases, and peroxidases. However, no significant presence of heavy metal resistance genes was detected in the *P. protegens* clade, as observed in the *P. aeruginosa* clade. Notably, the *P. protegens* E1BL2 strain exhibited genes involved in phosphorus and zinc mobilization (phosphatase, 3-phytase, zinc efflux transporter, zinc transporting ATPase) (Figure 2). 

Specifically, the genome analysis of *P. protegens* E1BL2 showed the putative genes involved in metabolite synthesis (*bdh*A, *aco*R, *trp*E, *spe*E, *pot*A); siderophores (*pta*A, *pch*C); and extracellular enzymes relevant for PGPB mechanisms (*cel*3, *chi*14), protection against oxidative stress (*hsc*A, *htp*G), nitrogen metabolism (*nir*D, *nit*1, *hmp*A), inductors of plant-induced systemic resistance (ISR) (*fla*A, *fla*G, *rff*A, *rfa*P), fungal biocontrol (*phl*D, *prt*D, *prn*D, *hcn*A-1), pest control (*vgr*G-1, *hig*B-2, *apr*E, *psl*A, *ppk*A), and the establishment of plant–bacteria symbiosis (*pga*A, *pga*B, *pga*C, *exb*D) (Table 1).

### 2.3. PGPB Characterization of P. protegens E1BL2

The *P. protegens* E1BL2 strain characterization focused on plant growth promotion characteristics (Table 2), which showed that this bacterium maintains some activities to help the plant obtain optimal development. The results of the IAA production, qualitative metallophore detection, and phosphate solubilization confirmed previously reported findings [24]. The strain solubilized 41.8 ± 1.6 µg/mL of insoluble phosphate to orthophosphate and produced 4.5 ± 0.7 µg/mL of IAA and metallophores for Fe^+3^, Mo^+6^, Zn^+2^, and V^+5^ ions, with percentages that ranged from 32 to 60%. Furthermore, the production in vitro of chitinases, proteases, and cellulases extracellular enzymes was detected. 

### 2.4. Determination of the Effect of Inoculation of Bacterial Strains in Maize Plants

Differences in the weights and lengths of the stems and roots of maize seedlings under different treatments were determined at 50 days in the greenhouse (Figure 3), where a growth promotion effect was observed in plants inoculated with endophytic bacteria. *P. protegens* E1BL2 significantly increased the weight and length of the majority of plants compared with the uninoculated control and the growth-positive control *Burkholderia metallica* R3J3HD10, which is a PGPB previously recognized in Conejo maize landrace plants [25]. 

### 2.5. Inhibition of Fungal Growth and Germination Protection In Vitro

In vitro, antifungal activity was evaluated by determining the percentage of inhibition of radial growth (PIRG) of phytopathogenic fungi grown in a PDA medium. The control 0.2% Cupravit^®^ solution presented 54.8 and 77.2% PIRGs for *Rhizoctonia* sp. and *Helminthosporium maydis*, respectively. The phytopathogenic fungi PIRGs of *P. protegens* E1BL2 were higher at 90.3, 73.0, 59.7, and 72.7% for *H. maydis*, *Pestalotia* sp., *Rhizoctonia* sp., and *Pythium* sp., respectively (Table 3).

Figure 4 presents the protection of the germination of Tuxpeño maize seeds by *P. protegens* E1BL2 based on the 18-point inhibition scale of the infection test. The treatments were compared against the seed infection of phytopathogenic fungi. *P. protegens* E1BL2 exhibited statistically significant protection of plants against all fungal infections.

### 2.6. Maize Protection Seedling Assay in Greenhouse

*P. protegens* E1BL2 + phytopathogenic fungi treatments showed an evident biocontrol effect under greenhouse conditions using native maize seedlings (Figure 5). Even the more susceptible maize landraces (Bofo and Tuxpeño) were partially protected from fungal attack by bacteria inoculation.

### 2.7. Plant Field Trial

In the field trials, the maize plants treated with *P. protegens* E1BL2 exhibited significantly greater plant heights and yields than those under conventional fertilization, growth media, and uninoculated control treatments. The statistical analysis using Tukey’s test revealed that the yield of *P. protegens* E1BL2-treated plants was significantly higher (* *p* ≤ 0.05) than that of the uninoculated controls. The visual assessment of maize cobs indicated that those from the *P. protegens* E1BL2 treatment were notably larger and more developed than those from the other treatments, which underscores the biofertilizer’s potential to enhance maize productivity and quality (Figure 6).

## 3. Discussion

This work explored the genomic and phenotypic traits of *P. protegens* E1BL2, which is a bacterium that was previously isolated from maize to confirm the presence of relevant genes and evaluate their associated phenotypic effects. The strain not only promoted plant growth but also exhibited biocontrol activity against phytopathogenic fungi and holds potential as a plant-growth-promoting bacterium (PGPB). Beyond its essential ecological and plant–bacteria symbiosis perspectives, this strain could contribute significantly to bioinoculant development. By formulating it with autochthonous bacteria, we can harness its diverse applications in agriculture, biotechnology, and environmental management to offer a promising future for these sectors. *P. protegens* E1BL2 is an endophytic bacterium isolated from the leaves of the Mexican giant maize Jala landrace and is part of a collection of 374 microorganisms isolated from this environment [24]. The strain stands out for its practical applications, such as improving maize seedlings’ development and growth and inhibiting the growth of phytopathogenic fungi, which instills confidence in its potential use in various fields.

The phylogenomic reconstruction of the *Pseudomonas* genus confirmed the seven main clades, tree topology, and species distribution previously recognized by previous works performed with 16S rRNA, *rpoD*, *rpoB*, and *gyrB* concatenated gene sequences [26,27,28,29,30]. The clade of *P. protegens* contains mainly plant rhizospheric or endophytic strains harboring important phenotypic features for establishing plant–bacteria relationships [31,32,33,34]. *P. protegens* harbors a collection of genes and genetic routes that encode known plant-growth-promoting molecules, such as acetoin [35], spermidine [36], and anthranilate [37], as well as metallophores [38,39,40], extracellular macromolecule-degrading enzymes [41,42,43], inducers of plant ISR of plants [44,45], and antimicrobials and enzymes involved in fungal biological control [23,36,42,46,47].

The symbiotic relationship between *P. protegens* E1BL2 and *Burkholderia metallica* R3J3HD10 is a complex interplay of mechanisms that increase the weight and length of most maize landraces’ roots and stems compared with an uninoculated control. This mutualistic symbiosis, which is broadly distributed in cultivated plants, such as grapevine, wheat, and maize, operates through intricate mechanisms, such as phosphate solubilization [48,49,50], ACC deamination [30,50], the production of IAA [48,50,51], and the secretion of metallophores that increase the bioavailability of some metallic cations essential for plant development [38,39,40,48,50]. However, the growth promotion of Bolita, Reventador, and Azul landraces was modest or undetected, which was likely due to an antagonistic effect of the high concentration of anthocyanins that characterize this native landrace against bacteria. Similar effects were observed with anthocyanins from other fruits, such as blueberries, blackberries, and grapes, where a high anthocyanin content was shown to inhibit bacterial growth [52,53].

The genotypic and phenotypic traits of *P. protegens* E1BL2 revealed diverse compounds with antimicrobial activity and inducers of plant ISR. The strain inhibited in vitro the growth of seven phytopathogen fungi. The antagonism of the strain against fungi is comparable and even, in some cases, more significant than the compound of copper oxychloride usually used as a pesticide [54]. Also, other *P. protegens* strains synthesize metabolites with antifungal activity, such as 2,4-DAPG (DAPG) [23,55,56,57], pyocyanin [56,58], pyrrolnitrin (PR) [18,56], pyoluteorin (PL) [56,57], orfamide A [41,56,59], rhizoxin A [56,60], cyanide acid (HCN) [56,61], and chitinases [41,42,62]. For example, in addition to its plant-growth-promoting abilities, *P. protegens* CHA0 isolated from roots of the tobacco plant can inhibit the oomycete *Pythium ultimatum*, *Fusarium oxysporum* basidiomycete, and even some herbivorous insects [26,63,64,65,66]. Also, *P. protegens* Pf-5 is a strain that is closely phylogenomically related with *P. protegens* E1BL2 and also inhibits the growth of phytopathogenic fungi through the production of DAPG, PR, PL, HCN, and rhizoxin [67]. Although no antifungal activities have been reported with *P. protegens* EMM-1, this strain isolated from the maize rhizosphere of Rojo Criollo landrace inhibits the growth of *Pseudomonas putida*, *Pseudomonas syringae*, and *Ralstonia solanacearum* phytopathogenic bacteria and *Klebsiella pneumoniae*, *Burkholderia cepacia* complex, and *Streptococcus* beta-hemolytic clinical pathogenic bacteria [68,69].

In plants, ISR is a defense mechanism activated by infection through several molecular stimuli that enhance the plant’s ability to resist pathogens. *P. protegens* strains express known inducers of plant ISR, such as flagellin [70,71], lipopolysaccharide [63,70], metallophores, and pyoverdines [64], which trigger a response aimed at limiting phytopathogenic fungi infection. Although genomic data of *P. protegens* E1BL2 do not directly demonstrate the ISR phenomenon, the strain genome harbors genetic routes that encode all the mentioned inductors of plant ISR.

The plant growth promotion, induction of plant ISR, and antifungal simultaneous capabilities of *P. protegens* were tested and proven in plant field trials using these bacteria as bioinoculants (Figure 7). In particular, some field trials were carried out with maize inoculated with different PGPB. A bioinoculant of *Azospirillum brasiliense* achieved grain yields between 6 and 9 tons/ha [72,73], while another bioinoculant containing *Bacillus subtilis* 160 reached 12.8 tons/ha [74]. Although it is not appropriate to compare field trials performed in different conditions, the maize productivity with a bioinoculant prepared with *P. protegens* Pf-5 X940 led to a substantial 115% increase in productive biomass [67]. Our field trial in this study reached an average yield of 16.6 tons/ha and 115.27% of productive biomass, which further validated the efficacy of *P. protegens* E1BL2 as a bioinoculant. 

There is not a large market for native maize landraces at the international level. However, in several Latin American countries, they are essential, both culturally and as food, in addition to contributing to the germplasm of the species. Mexico is the place of biological origin, with domestication from teosinte and further human-driven diversification of *Z. mays* [74,75,76]. Because of this, studying the microbiota associated with the maize races only found in Mexico, Guatemala, Peru, and other countries is crucial. Few studies explored the symbiosis of maize landraces with autochthonous culturable bacteria and their evolutionary, ecological, and biotechnological potential capabilities [25,76,77]. The symbiotic relationship between maize landraces and bacteria holds significant potential for biotechnological applications. For example, certain bacteria may enhance the plant’s nutrient uptake, which leads to increased crop yields. Other bacteria may produce antimicrobial compounds, which reduce the need for chemical pesticides. Because many maize landraces are more resistant to diseases and more tolerant to adverse environments than improved hybrid maize widely used in intensive agriculture [78,79], the search for improved varieties is still ongoing, and the understanding of the maize holobiont and its microbiome is beginning.

Our findings pave the way for innovative approaches in sustainable agriculture, which emphasize the importance of preserving and studying native maize landraces and their associated microbiota. This knowledge not only contributes to the conservation of biodiversity but also offers practical solutions for improving agricultural productivity and sustainability.

## 4. Materials and Methods

### 4.1. Bacterial Strain and Growth Conditions 

Bacterial endophyte *Pseudomonas protegens* E1BL2 were previously isolated from maize leaves of Jala landrace [25] and were routinely grown in R2A medium at 28 °C for 48 h.

### 4.2. DNA Extraction 

Genomic DNA extraction was performed following a CTAB-based protocol [80], which obtained the desired purity and integrity characteristics for complete genome sequencing.

### 4.3. Whole-Genome Sequencing and Assembly and Annotation 

The genomic libraries of *P. protegens* E1BL2 were sequenced and built using the Illumina HiSeq platform. The libraries were prepared with Nextera XT Library Prep, which leaves amplicons of 300 bp. The sequencing took place in Sacramento, CA, USA. The quality control of the raw data was analyzed with the FastQC v. 0.11.9 program [81]; internal quality control was also done by looking for the PhiX library, which derived from a small bacteriophage that had a size of approximately 500 bp with 45% GC (Illumina, San Diego, CA, USA). This control was performed because, in a standard run, 1% of PhiX was introduced. With the appropriate quality readings, the de novo assembly of the genome was performed with the SPAdes v. 3.13.0 program [82]. The quality evaluation of the assembly was performed with the QUAST v. 5.0.2 program [83]. Genome annotation was assigned with the online server RAST (Rapid Annotation using Subsystem Technology) v. 2.0 [84]. The search for biosynthetic clusters of secondary metabolites was carried out with the online server antiSMASH bacterial version v. 6.0 [85] and with PRISM (PRediction Informatics for Secondary Metabolomes) [86].

### 4.4. Phylogenomic and Comparative Genome Analysis

The taxonomic assignment of the E1BL2 strain was performed by phylogenomic analysis, including 39 strains that belonged to different species of the genus *Pseudomonas* using the M1CR0B14L1Z3R platform with a maximal e-value cutoff: 0.01, identity minimal percent cutoff: 90.0%, and minimal percentage for the core: 100.0% [87]. All genomes of the strains were recorded and compared to locate essential genes involved in plant–microorganism interaction, including the mobilization of phosphorus and zinc, heavy metal tolerance, phytohormones synthesis, biofilm formation, metallophores biosynthesis, biological control, quorum sensing, quorum quenching, peroxidases, catalases, superoxide dismutase, cellulases, and mechanisms to antibiotic resistance; the data were analyzed and visualized in TBtools-ll v1.108 [88].

### 4.5. PGPB Characterization of P. protegens E1BL2

The confirmatory determinations of the solubilization of phosphates, detection of the ACC deaminase enzyme, and production of indoleacetic acid and metallophores were determined according to the protocols previously reported [24]. In addition, the detection of extracellular enzymes, such as chitinases, cellulases, and proteases, was carried out using the protocols described [89].

### 4.6. Determination of the Effect of Inoculation of Bacterial Strains in Maize Plants

Germinated seeds were treated by seed-coating with bacterial suspensions (1 mL of 1 × 10^8^ CFU/mL per seed) and placed in pots with sterile vermiculite. The treatments included T1: uninoculated control (water only), T2: plant-growth positive control (*Burkholderia metallica* R3J3HD10), and T3: *Pseudomonas protegens* E1BL2. Each treatment was replicated six times. After 50 days, the plants were washed with water to remove excess vermiculite, and then the root and stem lengths and dry and wet weights were determined. The percentages of maize growth were calculated by taking the values of the uninoculated controls for each landrace in each set of treatments as a frame. Specifically, the value of the uninoculated controls for each maize landrace was taken as 100%, and values above and below this were scaled accordingly.

### 4.7. Inhibition of Fungal Growth and Germination Protection In Vitro

Plate inhibition assays of fungal growth were performed using a previously reported method on PDA media [89,90]. A commercial aqueous solution of 400 g/100 L copper chloride oxide (CUPRAVIT^®^, Bayer, Leverkusen, Germany) was used as a positive control for inhibition. As a treatment, a 1 × 10^8^ CFU/mL suspension of *P. protegens* E1BL2 was used. Three replicates of each sample were used in this assay. 

### 4.8. Maize Protection Seedling Assay in Greenhouse 

The Tuxpeño maize seeds were immersed in a suspension of bacteria adjusted at OD_600_ of 0.8 with agitation for 12 h at 28 °C. Control seeds were immersed in saline solution under the same conditions. Five seeds were placed on the surface of plates with 3% agar water. Subsequently, 100 μL of phytopathogenic fungi conidia (1 × 10^7^ conidia/mL) were inoculated on seeds and incubated at 25 °C for ten days. The inhibition of germination by phytopathogenic species was evaluated on the 18-point inhibition scale of the infection test previously described [91].

### 4.9. Test in Planta for Maize Protection Seedling at Greenhouse Level

The protection of the germination of Cacahuacintle, Elotes Cónicos, Bofo, and Tuxpeño maize landrace seeds by *Pseudomonas protegens* E1BL2 were evaluated against seven phytopathogenic fungi: *Fusarium oxysporum*, *Colletotrichum falcatum*, *Helminthosporium maydis*, *Curvularia* sp., *Pestalotia* sp., *Rhizoctonia* sp., and *Pythium* sp. A total of three PDA agar spots of 7 mm with mycelium growth of each phytopathogen fungi were transferred to sterile plastic pots containing 100 g of sterile ground corn hydrated with 40 mL of sterile water and incubated at 28 °C for ten days [91]. A total of 55 g of the corn/fungus mixture was weighed and added to a 1 kg pot that contained sterile vermiculite. The seeds were disinfected by soaking in 2% sodium hypochlorite for 10 min, followed by rinsing with sterile distilled water, and then germinated in Hoagland agar before inoculating the plants with bacteria. After three days of germination, the seeds were immersed for 20 min in a saline suspension with the bacterial biomass adjusted to an OD_600_ of 0.8 and were subsequently dried and planted in each pot. Three plants per pot were planted, and six replicates per treatment were assayed in the greenhouse of Plant Physiology at the National School of Biological Sciences, Mexico City, with photoperiods of 12 × 12 h.

The treatments for the four landraces were uninoculated control (water only), fungal infection control, strain E1BL2 safety control, and *P. protegens* E1BL2 + fungi. 

The plastic pots were placed at 28 °C with photoperiods of sunlight for 30 days. Stem and root heights were taken. Fresh and dry stem and root weights were also measured [92]. The symptoms of the infection were observed, and the percentage of the severity of the infection was evaluated by two scales: root and stem rot [93] and chlorosis, necrosis, and leaf spots [94].

### 4.10. Plant Field Trial

The plant field trial evaluated the *P. protegens* E1BL2 effect in the corn crop yield in the experimental fields “El Llano” from the Instituto Nacional de Investigaciones Forestales, Agrícolas y Pecuarias (INIFAP), Tula de Allende Hidalgo, México (20.0601722, −99.3012504). The sowing was undertaken in the third week of May 2022. The commercial hybrid Supremo from ASPROS was used for the trial. 

The seeds were extensively washed with tap water to remove the insecticides and fungicides, dried with absorbent paper, placed in a plastic bag, and inoculated with a suspension of *P. protegens E1BL2* at 1 × 10^8^ cells/mL with 30 mL/kg of seed. The bacteria were obtained from a culture of 48 h in Luria Bertani (LB) broth (Difco^TM^). Carboxymethylcellulose was used as an adherent for 30 g/kg of seed. The controls with distilled water and the other with Luria broth media were incorporated into the experiment. 

The evaluation was completely randomized in 6 repetitions in experimental plots of 5 m in length with four furrows with 80 cm of separation between each. Three seeds were sown at a separation of 17.5 cm, irrigated monthly, and Marvel^®^ herbicides (3,6-dichloro-2-methoxy benzoic acid and 6-chloro-N-ethyl-N-isopropyl-1,3,5-triazine-2,4-diamine) were dosed at 2 L/ha after the first irrigation. Later, at 30 days, a thinning was undertaken to obtain a uniform population of 112 plants per plot, which was equivalent to a population density of 72,500 plants/ha. The treatments employed included uninoculated control (no treatment), growth media (sterile LB culture medium), conventional fertilization (standard chemical fertilizer application), and *P. protegens* E1BL2 treatment (inoculation with *P. protegens* E1BL2 strain). The traditional treatment of fertilization involved applying a total of 200 kg/ha of nitrogen (N), 100 kg/ha of phosphorus (P_2_O_5_), and 50 kg/ha of potassium (K_2_O). Three applications of the bacteria (100 mL of 1 × 10^9^ CFU/mL per plant) were made monthly, started 25 days after sowing, and the products were applied manually via spraying.

Plant ears in the central rows were counted in each plot for yield determination, of which 22 were taken randomly. These were weighed and shelled. The kernels obtained were also weighed. Seed moisture was determined using a John Deere Brand Grain Moisture Tester. The shelled factor (SF) was calculated using the following formula:SF = (weight of kernels from 22 ears/total weight of 22 ears). 

With this data, the yield (Y) was determined using the following formula:Y = [(W)(T)(100-PG/86)(SF)(1000/G)]/1000 
where W—average weight of 22 ears (kg), T—total number of ears, HPG—humidity percentage of kernels, 86—standardized yield factor at 14% humidity, SF—shelled factor, and G—furrow width (0.8 m) [95]. 

### 4.11. Statistical Analysis

The data represent the arithmetic averages, and the error bars indicate the standard deviations. The results for the effect of inoculation, antifungal activity, germination protection, and protection at the greenhouse level were analyzed using a two-way ANOVA test followed by a Dunnett’s multiple comparison test using GraphPad Prism software version 9.0.1 for MacBook (GraphPad Software, La Jolla, CA, USA; https://www.graphpad.com, accessed on 1 May 2023). For the plant field trial, the analysis of variance (ANOVA) was made with a value of significance of 0.05; also, Tukey’s test was used to compare the yield of inoculated and uninoculated plants (* *p* ≤ 0.05) using the software Past 4.03.

## 5. Conclusions

This research explored the genomic content and phenotypic traits of *P. protegens* E1BL2 and its biological control capabilities. This study identified several genes and plant-growth-promoting and biocontrol activities that can be applied in agriculture, biotechnology, and environmental management. The research in vitro, greenhouse, and field trials also highlighted the potential of *P. protegens* as a biofertilizer, especially for crops like maize, due to its ability to solubilize phosphate and produce metallophores and IAA. Moreover, the findings demonstrated that *P. protegens* E1BL2 has the potential to control phytopathogenic fungi, which is probably a result of the production of secondary metabolites and the induction of systemic resistance in the plant. Therefore, this study demonstrated that native maize landraces harbor autochthonous PGPB, which can contribute to developing new eco-friendly bacterial bioinoculants for more sustainable agriculture.

## Figures and Tables

**Figure 1 ijms-25-09508-f001:**
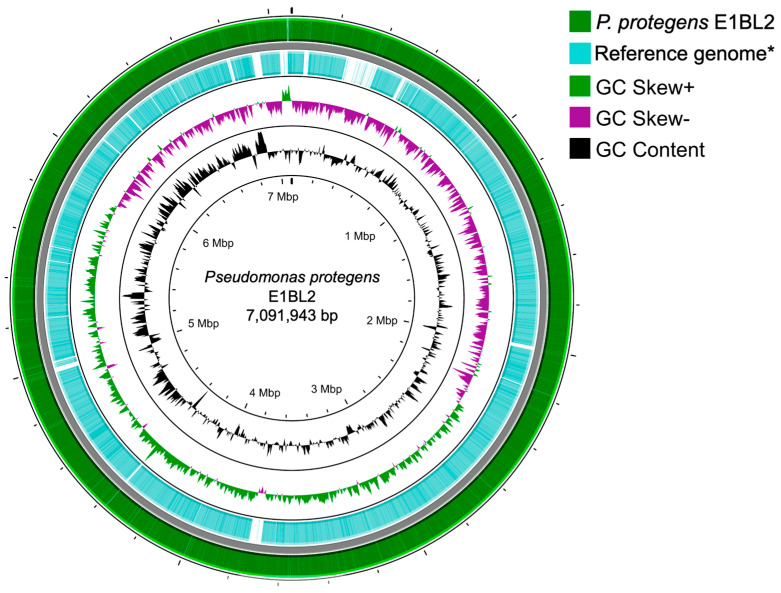
Representation of the circular genome of *Pseudomonas protegens* E1BL2. Using the Proksee server, the genome of *Pseudomonas protegens* E1BL2 was analyzed. The outermost circle represents the coding sequences of its genome assembly, which was deposited under GenBank BioProject PRJNA1078837, with BioSample ID SAMN40020743. The next circle contains the coding sequences of the *Pseudomonas protegens* SN15-2 * reference genome, which was phylogenetically the closest.

**Figure 2 ijms-25-09508-f002:**
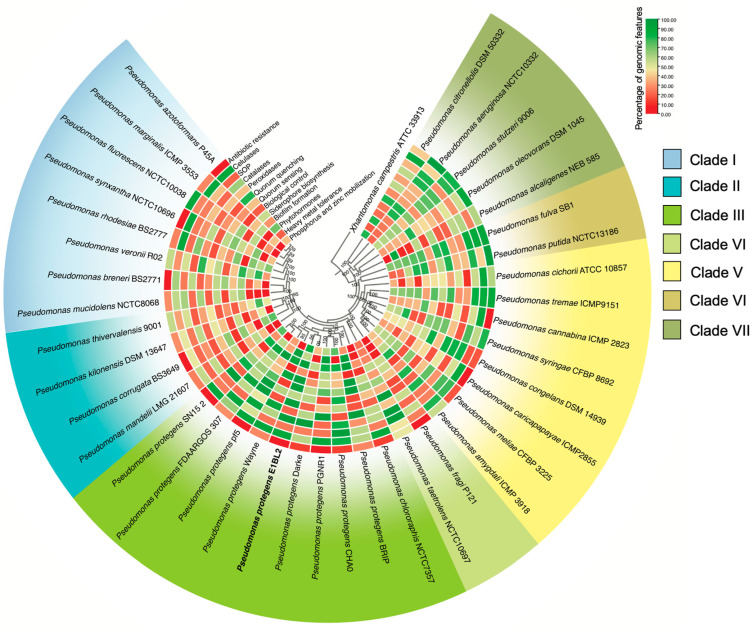
Phylogenomic tree and relevant plant-symbiosis genomic traits of *Pseudomonas protegens* strains and closely related species. The tree was built using the maximum likelihood method, which compared a core of 937 orthologous genes by employing 1000 bootstraps. Seven main clades were recognized. The color scale represents the percentage of genes found for each genomic trait for the 39 species analyzed; the red color represents a low percentage, while the green color represents a high percentage.

**Figure 3 ijms-25-09508-f003:**
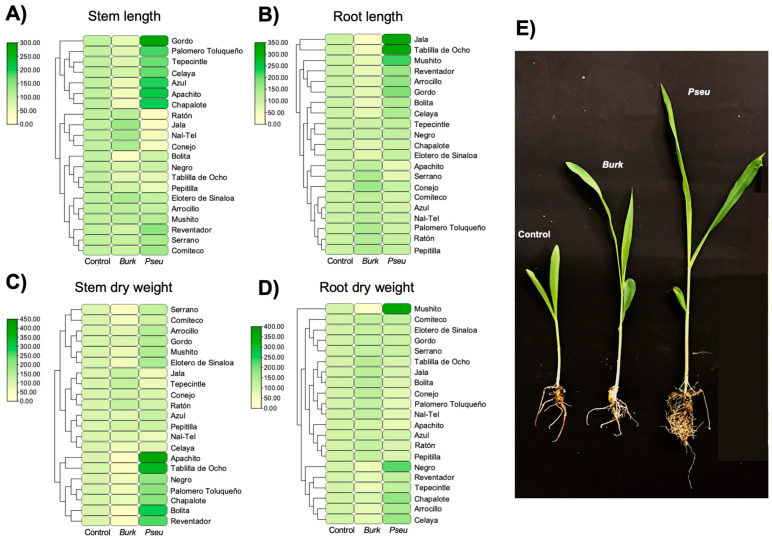
Growth percentage heatmap of various maize landraces inoculated with either *Burkholderia metallica* R3J3HD10 (*Burk*) or *Pseudomonas protegens* E1BL2 (*Pseu*). The growth parameters measured included the (**A**) stem length, (**B**) root length, (**C**) stem dry weight, and (**D**) root dry weight, with an uninoculated control as a negative control, *Burkholderia*-inoculated (*Burk*) as a positive control, and *Pseudomonas*-inoculated (*Pseu*) conditions. The color scale indicates the growth percentage, with green representing higher growth percentages and yellow representing lower growth percentages. The clustering dendrogram illustrates the similarity in growth responses between the different maize landraces. (**E**) Representative examples of treatments in Gordo maize landrace: uninoculated control, treatment with *Burkholderia*, and treatment with *Pseudomonas*.

**Figure 4 ijms-25-09508-f004:**
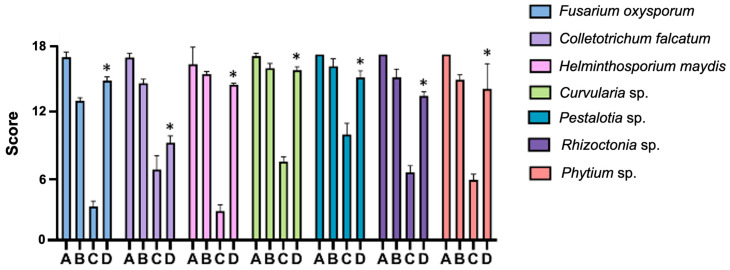
Protection of the germination of Tuxpeño maize seeds by *Pseudomonas protegens* E1BL2 against phytopathogenic fungi. *Fusarium oxysporum*, *Colletotrichum falcatum*, *Helminthosporium maydis*, *Curvularia* sp., *Pestalotia* sp., *Rhizoctonia* sp., and *Pythium* sp. were assayed. The 18-point inhibition scale of the infection test was used to evaluate the fungus damage to the seeds and protection assay by endophytic bacteria. A, uninoculated control; B, safety control of *P. protegens* E1BL2; C, fungal infection control; D, *P. protegens* E1BL2 + phytopathogenic fungi. Error bars represent standard deviations (SDs). Only statistically significant differences between infection control and *P. protegens* E1BL2 + phytopathogen fungi were determined using one-way ANOVA (* *p* < 0.05).

**Figure 5 ijms-25-09508-f005:**
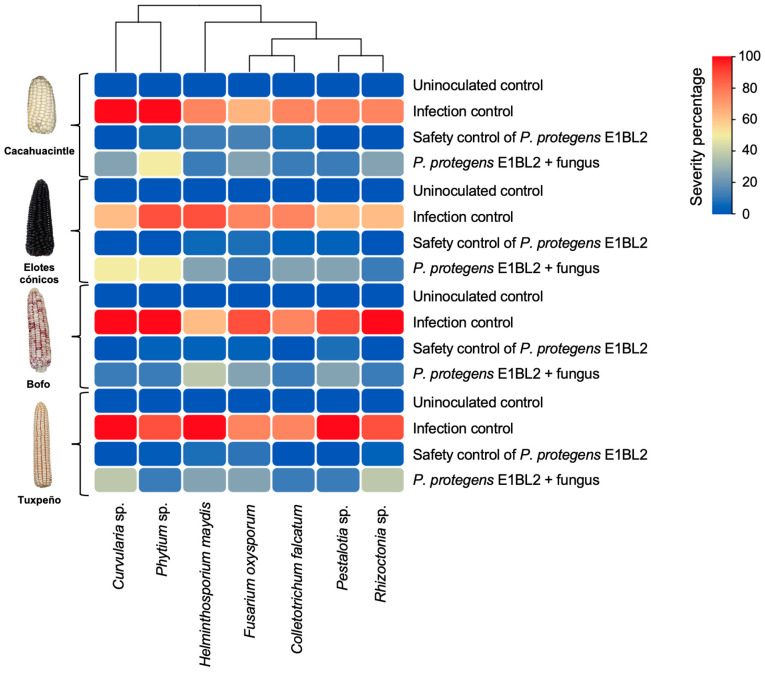
Mexican maize landrace biocontrol assays of *Pseudomonas protegens* E1BL2 against phytopathogenic fungi at the greenhouse level. The heat map shows the percentage of the infection severity of the seedlings during the different treatments. The red bars represent the most severe plant damage, while the blue bars represent the absence of or less damage.

**Figure 6 ijms-25-09508-f006:**
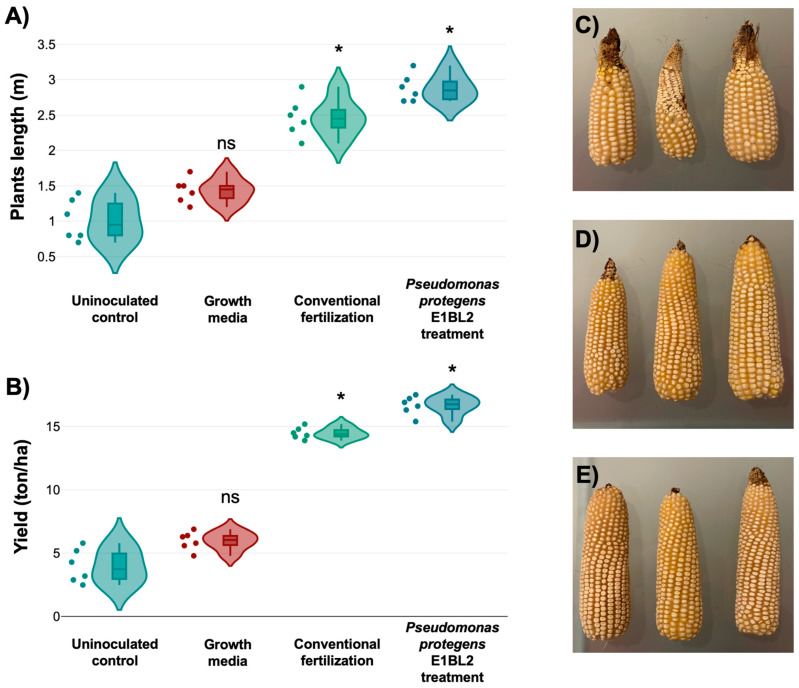
Plant field trial of *P. protegens* E1BL2 application in corn crop using the commercial maize hybrid Supremo. Panels (**A**,**B**) display violin plots illustrating the distribution of plant lengths (m) and yields (tons/ha), respectively, for treatments including uninoculated control, growth media, conventional fertilization, and *P. protegens* E1BL2 treatment, with statistically significant differences indicated by asterisks (* *p* < 0.05) and non-significant differences labeled as “ns”. Tukey’s test was used to compare the yields of inoculated and uninoculated plants (* *p* ≤ 0.05). (**C**) Uninoculated control, (**D**) conventional fertilization, and (**E**) *P. protegens* E1BL2 treatment.

**Figure 7 ijms-25-09508-f007:**
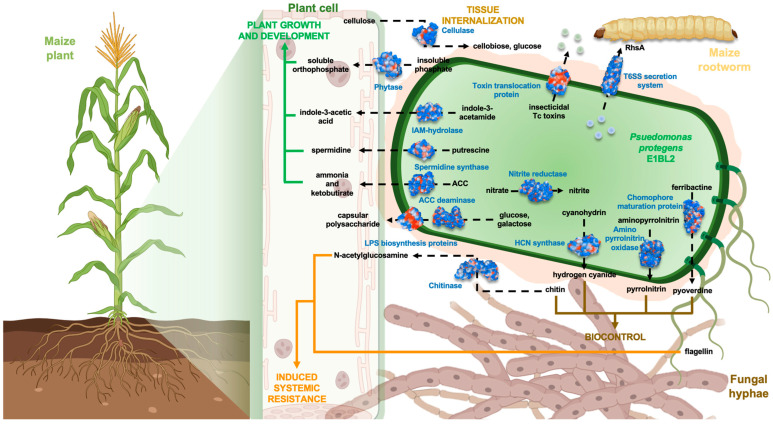
Graphical abstract of the *Pseudomonas protegens* E1BL2 functions in the PGPB–plant–phytopathogen compound interactions through phenotypic information and genomic traits. The image was made with BioRender.

**Table 1 ijms-25-09508-t001:** Relevant genes of *Pseudomonas protegens* E1BL2 in the plant–bacteria relationship.

Gen	Protein	Genomic Location
*bdhA*	2,3-butanediol dehydrogenase, R-alcohol forming, (R)- and (S)-acetoin-specific	173,948–175,051
*acoR*	Transcriptional activator of acetoin dehydrogenase	167,454–169,322
*trpE*	Anthranilate synthase, amidotransferase component	553,679–554,272
*speE*	Spermidine synthase	163,961–166,468
*potA*	Spermidine/putrescine import ABC transporter ATP-binding protein	114,777–115,868
*ptaA*	ABC transporter in pyoverdin gene cluster, periplasmic component	32,978–33,961
*pchC*	Pyochelin biosynthetic protein	903,310–904,089
*cel3*	Cellulase	191,379–192,467
*chi14*	Chitinase	34,754–36,220
*hscA*	Chaperon protein	498,330–500,192
*htpG*	Chaperon protein	189,071–190,975
*nirD*	Nitrite reductase	2–253
*nit1*	Nitrilase	870,762–871,667
hmpA	Nitric oxide dioxygenase	566,116–567,297
*flaA*	Flagellin protein	97,566–98,414
*flaG*	Flagellar protein	98,488–98,862
*rffA*	Lipopolysaccharide biosynthesis protein	94,647–95,207
*rfaP*	Lipopolysaccharide core heptose(I) kinase	33,553–34,359
*phlD*	Phloroglucinol synthase	17,378–19,240
*prtD*	Pyocin formation protein	538,488–539,531
*prnD*	Aminopyrrolnitrin oxidase	96,691–97,782
*hcnA*	Hydrogen cyanide synthase	265,107–265,424
*vgrG*	Translocation effector protein	130,901–132,085
*higB*	Toxin HigB	85,815–86,093
*aprE*	Type I secretion membrane fusion protein	595,315–596,649
*pslA*	Pellicle/biofilm biosynthesis protein	78,948–80,381
*ppkA*	T6SS Serine/threonine protein kinase	54,120–57,203
*pgaA*	Biofilm PGA outer membrane secretin	4930–7416
*pgaB*	Biofilm PGA synthesis deacetylase	2918–4915
*pgaC*	Biofilm PGA synthesis N-glycosyltransferase	1565–2914

**Table 2 ijms-25-09508-t002:** In vitro PGPB features of *Pseudomonas protegens* E1BL2.

PGPB Traits	Phosphate Solubilization		ACCd	IAA	Metallophores								Chitinase	Protease	Cellulase
					Fe^+3^		Mo^+6^		Zn^+2^		V^+5^				
	SQ	Qn	Ql	Qn	SQ	Qn	SQ	Qn	SQ	Qn	SQ	Qn	SQ	SQ	SQ
	(Index)	(µg/mL)	Nu	(µg/mL)	(Index)	(%)	(Index)	(%)	(Index)	(%)	(Index)	(%)	(Index)	(Index)	(Index)
*P. protegens* E1BL2	1.1 ± 0.3	41.8 ± 1.6	+	4.5 ± 0.7 *	2.0 ± 0.8	56 ± 3.6	3.0 ± 0.7	32 ± 3.5	2.1 ± 0.2	43 ± 4.2	4.4 ± 0.5 *	60 ± 3.9	2.7 ± 0.9	1.8 ± 0.3	1.6 ± 0.6 *
*B. metallica R3J3HD10*	2.0 ± 0.2	81.8 ± 3.6	-	2.7 ± 0.9	3.3 ± 0.5	62.4 ± 4.2	3.4 ± 0.4	40.2 ± 2.1	3.1 ± 0.8	46.7 ± 3.5	2.0 ± 0.6	20.3 ± 3.4	2.1 ± 0.1	2.6 ± 0.2	0.6 ± 0.1

Ql, qualitative; SQ, semi-quantitative; Qn, quantitative; Nu, no units; statistically significant differences indicated by asterisks (* *p* < 0.05).

**Table 3 ijms-25-09508-t003:** Percentage of inhibition of radial growth of phytopathogenic fungi by endophytic *Pseudomonas protegens* E1BL2 in dual cultures.

	*Fusarium oxysporum*	*Colletotrichum falcatum*	*Helminthosporium maydis*	*Curvularia* sp.	*Pestalotia* sp.	*Rhizoctonia* sp.	*Pythium* sp.
Cupravit^®^ (copper oxychloride)	7	58.5 ± 2.3	77.2 ± 1.8	69.1 ± 0.7	72.7 ± 3.2	54.8 ± 2.7	69.4 ± 1.3
*Pseudomonas protegens* E1BL2	63.8 ± 0.4 *	41.0 ± 2.4 *	90.3 ± 2.1 *	40.2 ± 5.0 *	73.0 ± 4.1	59.7 ± 6.5	72.7 ± 5.4

The percentages of inhibition of the radial growth of each of the fungi are shown, taking into account the radius from the center of the fungal colony to the periphery of both the spot zone of the bacteria/antifungal and the spot-free zone; only statistically significant differences between the infection control and *P. protegens* E1BL2 and Cupravit control were determined using Tukey test (* *p* < 0.05).

## Data Availability

The *Pseudomonas protegens* E1BL2 genome sequence herein generated was deposited in the GenBank BioProject (PRJNA1078837) with genome assembly and BioSample ID (SAMN40020743).

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
