# Peer review of "Genomic Insights into Pseudomonas protegens E1BL2 from Giant Jala Maize: A Novel Bioresource for Sustainable Agriculture and Efficient Management of Fungal Phytopathogens"

_ijms, 2024, doi:10.3390/ijms25179508_

Round 1

Reviewer 1 Report (New Reviewer)

Comments and Suggestions for Authors

a thorough paper     which reads well  thank you 

I asked for more detail in many places  especially methods and details on replication for stat analysis --  these are simple fixes  

not sure of the validity of the seed germination study think it is problematical

but the field study and lab study OK   

Author Response

Independent Review Report, Reviewer 1

EVALUATION
First, the author thanks Reviewer 1 for their insightful feedback on our research article, this work improved substantially with their contributions and suggestions.

Reviewer 1. also pollute run-off waters

Authors' response. In line 35 added the sentence “pollution of run off waters”.

Reviewer 1. Certain of these micro...

Authors' response. In line 36 added the sentence “Certain of these microorganisms”.

Reviewer 1. You have two various

Authors' response. In lines 44-46, we changed the sentence to “Many of the plant growth-promoting bacteria (PGPB) also enhance plant health through various mechanisms.”

Reviewer 1. Are you meaning siderophores?

Authors' response. The term siderophores refers to molecules capable of chelating iron, although some of these molecules can chelate other types of ions, we consider that it would be appropriate to call them metallophores.

Reviewer 1. Split into two sentences

Authors' response. In lines 49-53 we split the sentence to: “Additionally, PGPB produces enzymes such as cellulases and amylases, facilitating nutrient release from organic matter in the soil [4]. Furthermore, they synthesize secondary metabolites such as 2,4-diacetyl phloroglucinol (DAPG), pyoverdine (PV), pyrrolnitrin (PR), pyoluteorin (PL), pyocyanin (PC), orphamide A, hydrogen cyanide (HCN), chitinases, which act as a biocontrol mechanism [5].”

Reviewer 1. and

Authors' response. In line 69 we added the word “and”

Reviewer 1. Some chitinases plural

Authors' response. In lines 74-75, we changed the sentence to “some hydrolytic chitinases enzymes”.

Reviewer 1. A? does not make sense with The

Authors' response. In line 78, we changed the word to “A”.

Reviewer 1. bacterium

Authors' response. In line 89 we changed the word to “bacterium”.

Reviewer 1. Genes pertaining to strategies

Authors' response. In lines 90-91, we changed the sentence to “The draft genome harbored many recognized genes associated with strategies to promote plant growth and suppress the development of phytopathogenic fungi”.

Reviewer 1. Is this a maize colonist? I am more familiar with Pf5

Authors' response. Strain SN15-2 was isolated from the rhizosphere of tomato, but at the genome level, our strain has more similarity than the Pf5 strain.

Reviewer 1. Any plasmids?

Authors' response. Our genomic scan did not detect plasmid material.

Reviewer 1. Can you define more here?

Authors' response. In lines 120-121, we added the sentence “(phosphatase, 3-phytase, zinc efflux transporter, zinc transporting ATPase)”.

Reviewer 1. What is PGA?

Authors' response. It refers to the name of the gene pgaB and means poly-β-1,6-N-acetylglucosamine.

Reviewer 1. What do you mean by % do not understand a single siderophore eg pyoverdine type will bind metal ions differentially

Authors' response. The chelation percentage was calculated as described in the cited paper in the Materials and Methods section, using the following formula: Chelation percentage = [(AR − AS)/AR] × 100, where AR is the absorbance of the reference (CAS media without inoculation) and AS is the absorbance of the test sample (CAS media with bacterial inoculation).

Reviewer 1. These are different metal chelators?

Authors' response. We used CAS media with various ionic salts of different oxidation states, suggesting that the metal chelators may differ accordingly.

Reviewer 1. What is index?

Authors' response. The index is the product of the dividing halo and colony diameters in solid media, it is also cited in the Materials and Methods section.

Reviewer 1. What age of culture, what medium, etc

Authors' response. The details of media and time used are in the Materials and Methods section.

Reviewer 1. What was growth medium? what was control? how was inocula added? Did you confirm colonization of plant as epiphyte? or as endophyte?

Authors' response. Please refer to section 4.6 for the details you requested. We confirmed endophytic colonization in a previous study, so we did not consider it necessary to include it again in this manuscript.

Reviewer 1. Where is this microbe found in planta?

Authors' response. In line 162 we added the [25] reference that mentions the origin of the Burkholderia strain.

Reviewer 1. Solid or liquid? Replications? What is this? Explain it is a Cu fungicide commercial

Authors' response. Please refer to section 4.7 for the details you requested.

Reviewer 1. Is 54.8 statistically different than 59.7?

Authors' response. In Table 3, an asterisk (*) was added, and a clarification was included in lines 183-184: 'Statistically significant differences between the infection control, P. protegens E1BL2, and Cupravit control were determined using the Tukey test (p < 0.05)”.

Reviewer 1. What do you mean protection of germination? I do not believe all of these pathogens are in-the-field germination inhibitors. So B has no added pathogens. Please clarify.

Authors' response. In Figure 4, we included an uninoculated control and an infection control which showed a significant reduction in the germination across all fungi tested. We used the 18-point scale proposed by Borah et al., 2016, as it evaluates not only germination but also signs and symptoms of seed damage caused by the fungi. Germination protection refers to the effect of the Pseudomonas strain in preventing fungal damage during seed germination. The B) or safety control was to verify that the Pseudomonas strain did not affect germination and that when the bacteria were combined with the fungus, no confounding results would occur.

Reviewer 1. pathogenic fungi?

Authors' response. In line 199 the term “phytopathogenic fungi” was added.

Reviewer 1. Again wish the methods were earlier no idea what was done at this point

Authors' response. We apologize for the confusion. However, due to the template format, the results and discussion are presented first, followed by the materials and methods section.

Reviewer 1. This is all the way to harvest?  or just biomass?

Authors' response. For the field trials, the measurements were taken all the way to harvest, which is why we reported yield instead of weight.

Reviewer 1. Was this nutrition based though or stimulation of plant resistance? this is inorganic salts or compost  or both, what do you mean by growth media?

Authors' response. Please refer to section 4.10 for the details you requested.

Reviewer 1. ABCDE labelling could be added to each figure to help the reader understand treatment, did you have corn ear worm problems? or is natural senescence at the corn tip  usual for this variety, its a dry maize not  sweet corn?

Authors' response. Labels were added, and each one is clarified in the figure caption. The tip senescence is common in this corn variety, indicating a mature, fully dried ear.

Reviewer 1. gene products were not always measured

Authors' response. In lines 243-245 the sentence: “This work delves into the genome and expressed phenotypic traits of P. protegens E1BL2, a bacterium previously isolated from maize” was changed to “This work explores the genome and phenotypic traits of P. protegens E1BL2, a bacterium previously isolated from maize, confirming the presence of relevant genes and evaluating their associated phenotypic effects”.

Reviewer 1. Again you use other terms so  define  or change for consistency

Authors' response. We changed the term “Siderophores” to “Metallophores” in lines 85, 264, 273, 299, 366, 372, and 472.

Reviewer 1. explain all terms

Authors' response. In lines 285-287 the terms mentioned further below are abbreviated.

Reviewer 1. Again what medium used?

Authors' response. In line 390 the “on PDA media” sentence was added.

Reviewer 1. How disenfested?

Authors' response. In lines 410-412, the “The seeds were disinfected by soaking in 2% sodium hypochlorite for 10 minutes, followed by rinsing with sterile distilled water, then germinated in Hoagland agar before inoculating the plants with bacteria” sentence was added.

Reviewer 1. Why and how this seems a problem control?

Authors' response. The growth media control was used to ensure that the observed effects were due to the interaction of the microorganism itself and not the culture media in which it was grown, as remnants of the media may still be present when the bacteria are applied in the field.

Reviewer 2 Report (New Reviewer)

Comments and Suggestions for Authors

The manuscript submitted by Vega-Camarillo et al. reported the genomic information of P. protegens E1BL2, and characterized it as PGPB and inhibition fungi infection. In general, it is intresting for the readers. However, several issues should be further modified before formal publication. The detail comments refer to the attachment file.

Author Response

Independent Review Report, Reviewer 2

EVALUATION
First, the author thanks Reviewer 2 for their insightful feedback on our research article, this work improved substantially with their contributions and suggestions.

Reviewer 1. If the phenotype of promoting growth, plus the phenotype of promoting growth as the first result of the paper.

Authors' response. Thank you for your observation. We understand the importance of clearly distinguishing between the different growth-promoting phenotypes discussed in the paper. We prioritized the genomic analysis because we believe it is a novel and important tool that complements the phenotypic analyses that have been conducted over the years.

Reviewer 1. Which was the control or list the reasonable range of each assay? please display the control and give statistic analysis to show the prominent function on promoting plant growth.

Authors' response. In Table 2, the values obtained with the control strain are included, and an asterisk marks the tests where Pseudomonas showed statistically significant improvements.

Reviewer 1. Replace the panel letters a), b), c) in FIg. 3E with control, Burk and Pseu, respectively. It will be clearer and conciser.

Authors' response. In Figure 3 the letters a), b), and c) were replaced by “Control, Burk, and Pseu” words

Reviewer 1. How many times repeat? and how many samples for each treatment?

Authors' response. In lines 399-400 the sentence: “Three replicates of each sample were used in this assay” was added.

Reviewer 1. What was Score in vertical coordinates behalf of? why not use the percent of germination which can be direct standard?

Authors' response. We used the 18-point scale proposed by Borah et al., 2016, as it evaluates not only germination but also signs and symptoms of seed damage caused by the fungi.

Reviewer 1. most

Authors' response. In line 216 the word “most” was added.

Reviewer 1. blue

Authors' response. In line 217 the word “green” was changed to “blue”.

Reviewer 1. Present the different treatments in panel C, D and E.

Authors' response. The descriptions for panels C), D), and E) are provided in the figure legend in lines 239-240 to avoid obscuring the image with text.

Reviewer 1. The result describing should be put in the Result section of text instead of in Figure legend.

Authors' response. The figure legend for Figure 6 was revised to remove the description of the results, as they were already detailed in the appropriate section.

This manuscript is a resubmission of an earlier submission. The following is a list of the peer review reports and author responses from that submission.

Round 1

Reviewer 1 Report

Comments and Suggestions for Authors

General information:

This is a potentially interesting manuscript presenting the genome analysis of Pseudomonas protegens E1BL2. However, I recommend that this manuscript should undergo major revisions before it can be accepted for publication in IJMS. The language of the manuscript has to be carefully checked sometimes the manuscript is hard to follow even the abstract, which could discourage potential readers. The provided introduction is insufficient. Pseudomonas is a wide and diverse genus of bacteria that harbors stains pathogenic to plants and animals but also numerous plant-beneficial strains and even closely related strains are isolated from diverse environments. This should be reflected in the approach to genome analysis which should also include looking for potential toxins or human pathogenicity factors. This genus is very well studied which should be reflected in the introduction. I have added a suggested citation. Please find a related literature review and try to revolve your introduction around it. Currently given information is very general and does not reflect the scope of the presented research. It is unclear to me if the genome sequencing is part of this manuscript. If yes more information should be included as supplementary material if not the publication should refer to the study presenting the genome. The results are not clearly presented and are hard to follow, due to language mistakes. When I try to understand the results and go to appropriate methods, I find that the methods do not clearly describe the used method as well. I understand that some methods were prepared according to other manuscripts but short information should be included here as well. These are not well-established methods known by everyone and many versions of these methods are used. Also in the explanation of each method, the information on the number of repetitions method of data collection and duration should always be included. All experiments should have appropriate controls including phenotypic analysis. Please remove the empty pages from the manuscript. In the discussion there is brief information about the symbiotic interaction with B. metallica R3J3HD10 which has not been anyhow analyzed in the manuscript, please either remove that information as irrelevant or present it in a way that corresponds to the presented results. The figures included in the manuscript are very esthetic and detailed and are the strong side of this article. Although I would recommend extending the figure captions. Please also confirm that your Bio render license allows you to use the figures in Open Access publications, as it sometimes can be an issue. The discussion of the manuscript should concentrate more on presented results, and compare them to other results from similar experiments. Although in my opinion the introduction and discussion are slightly oversimplified the literature includes relevant and up-to-date studies. Therefore I would advise using one or two review papers to help organize the introduction and discussion, in a way that will be more clear and present a storyline leading to conclusions. Concluding this can be an interesting manuscript, but should undergo major revisions, especially concerning the presentation and language. I hope my following comments will aid the authors in this task.

In-text comments:

Line 21: Be more precise

Line 35: protect crops from

Line 36: soil erosion is not directly caused by pesticides, pesticides can reduce microbial diversity, increase the resistance of pathogens to pesticides, and cause soil pollution, especially in the case of broad-spectrum chemicals.

Line 47: try to separate the mechanisms by the modes of action, production of cellulases may be important for PGPRs but if you put it just beside plant hormones it looks somewhat strange. Please explain shortly the mechanisms to present this metabolite function in plant growth promotion

Line 51: This is not a sentence, please rephrase

Line 59: Please rephrase this paragraph, Pseudomonas is a diverse genus of bacteria with many strains that find use in biological plant protection but also many plant and animal pathogenic strains. Please add some citations regarding Pseudomonas in biocontrol there are many interesting reviews on that subject from renowned scientists including but not limited to: Loper et al 2012 (e1002784. https://doi.org/10.1371/journal.pgen.1002784. A) please free to add more recent papers since much has been discovered in that subject since that time. During Pseudomonas strain genome analysis it is important to look for genes that can be responsible for human pathogenicity, which can limit the application of such strains.

Line 102: Please write what program was used to create this graphic and used genome accession numbers.

Line 115: Unclear what does it mean, do you mean that P protegens have a higher number of genes responsible for phosphate metabolism than other species from the genus?

Line 148: Please compare the activities of the tested strain to at least one more strain preferably the type strain of the species.

Line 156: Please choose a different system of data presentation, this is strongly misleading, why does negative control always grow to 50% of plant growth? If you wish to present data compared to the control nor row results then compare to the control so if the average shoot length is 10cm and the average treatment control would be 15cm the treatment would give 150% or compare to positive control in this case if positive control would be 20cm then treatment would give 75% go positive control growth promotion.

Line 184: What is a safety control

Line 187: Please do not use a parametric test for non-parametric data.

Line 189: Apparent??

Line202: Fungus -> pathogen

Line 216: There is some problem with line numbering

Line 217: It is unclear what statistics are presented in panel B and D

Line 228 bacterium

Line 248: This is the first mention of that,

Line 280 strains not species

Line 284 Unclear how could bacterial genome demonstrate that?

Line 299: Please confirm to the associate editor that you have a license for the use of this program in publications.

Line 324: The genome was sequenced for this manuscript if yes please give more information as supplementary materials if not provide an appropriate citation.

Line 356: How roots were measured, were they washed, and if yes how?

Line 374: What was bacterial cfu?

Line 395: photoperiods of sunlight: please provide the site localization and timing of the experiment along with the length of the day during that period.

Line 399: Where can I find this data?

Line 409: LB is an abbreviation of Lysogenic Broth MacWilliams, M. P., & Liao, M. K. (2006). Luria broth (LB) and Luria agar (LA) media and their uses protocol. ASM MicrobeLibrary. American Society for Microbiology2006.

Comments on the Quality of English Language

Language must be improved some parts are hard to follow. Especially, please give the abstract special attention as it reflects on the delivery of the whole manuscript.

Reviewer 2 Report

Comments and Suggestions for Authors

This manuscript presents a comprehensive study on the genomic and phenotypic characteristics of the endophytic bacterium Pseudomonas protegens E1BL2, isolated from giant Jala maize. The research highlights the potential of this bacterium in promoting plant growth and controlling phytopathogenic fungi. The manuscript is well-structured, data-rich, and the experimental design is robust, making it a valuable contribution to sustainable agricultural practices. However, there are several areas that require revision for clarity and precision.

Title:

Suggestion: "Genomic Insights into Endophytic Pseudomonas protegens E1BL2 from Jala Maize: A Bioresource for Sustainable Agriculture and Fungal Pathogen Management."

Abstract:

The abstract is comprehensive but could be more concise, focusing on key results and implications.

Suggestion: "This study explores the relationship between plants and bacteria in agroecosystems, focusing on the endophytic Pseudomonas protegens E1BL2 isolated from Jala maize. We evaluated its plant growth promotion and biocontrol capabilities against eight phytopathogenic fungi under in vitro and greenhouse conditions, and in a field trial with hybrid maize CPL9105W. Results showed increased grain productivity and reduced rust incidence. Genomic analysis revealed genes involved in metabolite synthesis, siderophore production, oxidative stress protection, nitrogen metabolism, ISR induction, fungal biocontrol, pest control, and symbiosis establishment. Our findings suggest that P. protegens E1BL2 significantly promotes maize growth and offers biocontrol benefits, highlighting its potential as a bioinoculant."

Introduction:

Suggestion: "In the current biotechnological era, there is an ongoing effort to improve agricultural practices by moving away from external agrochemicals and broad-spectrum antimicrobials, which, while protective against pests and pathogens, have led to soil erosion and the displacement of beneficial rhizospheric and endophytic microorganisms. These microorganisms are crucial for promoting plant growth and protecting plant tissues by inhibiting phytopathogen proliferation."

Methods:

Suggestion: "Germinated seeds were treated by seed-coating with bacterial suspensions (1 mL of 1x10^8 CFU/mL per seed) and placed in pots with sterile vermiculite. Treatments included: T1: Uninoculated control (water only), T2: Plant-growth positive control (Burkholderia metallica R3J3HD10), and T3: Pseudomonas protegens E1BL2. Each treatment was replicated six times. After 50 days, root and stem lengths, as well as dry and wet weights, were measured."

Discussion:

Emphasize the broader implications of your findings for sustainable agriculture and biocontrol strategies. Highlight any potential limitations and future research directions.

Minor Edits:

Ensure all acronyms are defined upon first use.

Double-check for grammatical errors and typos.

Verify that all references are correctly formatted according to the journal’s guidelines.

This manuscript presents significant findings that contribute to our understanding of Pseudomonas protegens E1BL2's role in plant growth promotion and biocontrol. With the suggested revisions, it will provide even clearer and more impactful insights for the scientific community.I recommend this manuscript for publication after the above revisions are made.

Comments on the Quality of English Language

Minor editing of English language required

Round 2

Reviewer 1 Report

Comments and Suggestions for Authors

My major concern about this research is what is the topic of the presented study. Since most of the presented study seem to be already been published in Frontiers Microbiology:

De la Vega-Camarillo E, Sotelo-Aguilar J, Rios-Galicia B, Mercado-Flores Y, Arteaga-Garibay R, Villa-Tanaca L, Hernández-Rodríguez C. Promotion of the growth and yield of Zea mays by synthetic microbial communities from Jala maize. Front Microbiol. 2023 May 19;14:1167839. doi: 10.3389/fmicb.2023.1167839. PMID: 37275168; PMCID: PMC10235630.

I understand that sometimes especially the results from field experiments can be published in different manuscripts but they should always have appropriate controls. Unfortunately, I cannot find the genome used in your study and it is not clear to me if this publication is supposed to be a genome announcement or genome analysis. This should be stated clearly. I still miss detailed information on the used methodology. I am afraid that in the current state the manuscript is not understandable and I recommend rejecting this article and encouraging resubmission after major revisions.

Line 2: Please do not divide words in the title

Line 32: Please divide this sentence and avoid using such long sentences.

Line 37: Which practices do you refer to?

Line 47: remove essential

Line 50 PGPB produce (bacteria are plural)

Line 123: Again it is doubtful that bacteria species exist without some genes responsible for phosphate mobilization

Comments on the Quality of English Language

Some parts of the manuscript have been improved, but I still miss the main idea of this manuscript. The language should be more consistent, and sentences shorter.
